# Overexpression of Mitochondrial IF1 Prevents Metastatic Disease of Colorectal Cancer by Enhancing Anoikis and Tumor Infiltration of NK Cells

**DOI:** 10.3390/cancers12010022

**Published:** 2019-12-19

**Authors:** Lucía González-Llorente, Fulvio Santacatterina, Ana García-Aguilar, Cristina Nuevo-Tapioles, Sara González-García, Zuzana Tirpakova, María Luisa Toribio, José M. Cuezva

**Affiliations:** 1Departamento de Biología Molecular, Centro de Biología Molecular Severo Ochoa, Consejo Superior de Investigaciones Científicas-Universidad Autónoma de Madrid (CSIC-UAM), 28049 Madrid, Spain; lucia.ovi@gmail.com (L.G.-L.); fsantacatterina@cbm.csic.es (F.S.); ana.garcia@cbm.csic.es (A.G.-A.); cnuevo@cbm.csic.es (C.N.-T.); sgonzalez@cbm.csic.es (S.G.-G.); zuzana.tirpak@gmail.com (Z.T.); mtoribio@cbm.csic.es (M.L.T.); 2Centro de Investigación Biomédica en Red de Enfermedades Raras (CIBERER), ISCIII, 28049 Madrid, Spain; 3Instituto de Investigación Hospital 12 de Octubre, Universidad Autónoma de Madrid, 28049 Madrid, Spain

**Keywords:** colorectal cancer, ATPase Inhibitor Factor 1, metastasis, immune surveillance, prognosis

## Abstract

Increasing evidences show that the ATPase Inhibitory Factor 1 (IF1), the physiological inhibitor of the ATP synthase, is overexpressed in a large number of carcinomas contributing to metabolic reprogramming and cancer progression. Herein, we show that in contrast to the findings in other carcinomas, the overexpression of IF1 in a cohort of colorectal carcinomas (CRC) predicts less chances of disease recurrence, IF1 being an independent predictor of survival. Bioinformatic and gene expression analyses of the transcriptome of colon cancer cells with differential expression of IF1 indicate that cells overexpressing IF1 display a less aggressive behavior than IF1 silenced (shIF1) cells. Proteomic and functional in vitro migration and invasion assays confirmed the higher tumorigenic potential of shIF1 cells. Moreover, shIF1 cells have increased in vivo metastatic potential. The higher metastatic potential of shIF1 cells relies on increased cFLIP-mediated resistance to undergo anoikis after cell detachment. Furthermore, tumor spheroids of shIF1 cells have an increased ability to escape from immune surveillance by NK cells. Altogether, the results reveal that the overexpression of IF1 acts as a tumor suppressor in CRC with an important anti-metastatic role, thus supporting IF1 as a potential therapeutic target in CRC.

## 1. Introduction

Colorectal cancer is the third most common type of cancer worldwide [1] and the need for biomarkers that inform of disease progression is required. A large number of studies have highlighted the importance of metabolism in cancer initiation and progression [2,3]. Mitochondria play a central role in intermediary metabolism, the provision of metabolic energy and in cellular signaling [4,5]. A primary hub of mitochondrial activities is the ATP synthase, the engine of oxidative phosphorylation that utilizes the proton electrochemical gradient generated by the respiratory chain for the synthesis of ATP [4]. Inhibition of the ATP synthase results in the inhibition of mitochondrial respiration and the concurrent activation of aerobic glycolysis, which is a hallmark of proliferation in normal, cancer and stem cells [4,6,7,8]. An enhanced glycolytic flux provides the carbon skeletons and reductive power required for the biosynthesis of amino acids, lipids, nucleotides and energy that are necessary to sustain proliferation [9,10].

The ATPase Inhibitor Factor 1 (IF1) is a physiological inhibitor [11,12,13] and structural organizer [14] of the mitochondrial ATP synthase in tissues where it is normally expressed [15], as recently stressed in vivo [16,17,18]. IF1 is expressed at low levels in normal human colon, lung and breast tissues [15] but is highly overexpressed in the carcinomas arising in these tissues [13]. Moreover, the activity of IF1 as inhibitor of the ATP synthase is regulated by phosphorylation [19]. Dephosphorylated IF1 binds and inhibits the activity of the enzyme in hypoxia and during progression through the cell cycle whereas phosphorylated IF1 is unable to bind the enzyme and allows ATP synthesis upon an increase in energy demand in cardiac muscle in vivo [19]. In colon, lung and breast carcinomas, IF1 is largely present in its dephosphorylated state and hence active as an inhibitor of the ATP synthase [19]. Consistently, a high expression level of IF1 in human hepatocarcinomas [20], bladder [21] and stomach [22] carcinomas and in gliomas [23] is associated with a worse overall and/or disease free survival of the patients. In agreement with the pro-oncogenic role of IF1, we have observed that diethylnitrosamine-induced hepatocarcinogenesis in mice that overexpress an active IF1 mutant in the liver develop more and bigger tumors than control mice [16]. Mechanistically, the overexpression of IF1 in human hepatocarcinomas [20] activates the NFκB pathway and primes metastatic disease through Snai1- and VEGF-guided epithelial to mesenchymal transition (EMT) and angiogenesis, respectively.

Although the pro-oncogenic role of IF1 is supported in a large set of human cancers, its high expression level in the carcinomas is not always associated with a poorer patient outcome. In fact, disease free survival of breast cancer patients is significantly extended when the carcinomas overexpress IF1 [13,24]. Moreover, evidence from the ongoing study of a cohort of colon cancer patients indicated that a high expression of IF1 in the carcinomas also predicts a better overall survival for the patients [13]. Altogether, the purpose of this study was to deepen into the significance of IF1 as a biomarker of colorectal carcinomas (CRC) prognosis and to investigate the mechanisms underlying the role of IF1 in colon cancer progression. The findings in the cohort of CRC patients studied indicate that the tumor content of IF1 is an independent predictor of survival, advancing that a high tumor content of IF1 predicts longer time-periods for disease recurrence. Mechanistically, the results support that the overexpression of IF1 in colon cancer cells prevents metastatic disease by favoring anoikis and immune surveillance of the tumors by NK cells of the innate immune system.

## 2. Results

### 2.1. Increased Expression of IF1 Is Associated with Increased Survival of CRC Patients

Figure 1A shows representative images of the immunostaining of IF1 in normal and tumor tissue of colon cancer patients to illustrate the increase in IF1 expression triggered by carcinogenesis. Recombinant human IF1 (r-IF1) [12] was used to estimate the approximate amount of IF1 present in normal and tumor tissue of cancer patients (Figure 1B and Appendix A). The plot of the linear increased in signal when the amount of r-IF1 increases (Figure 1B and Appendix A) was used to interpolate the values of native IF1 signals in the samples (Figure 1B and Appendix A). The results indicate that the tumor content of IF1 roughly increased by a factor of two when compared to the adjacent normal tissue (Figure 1B and Appendix A). A similar two-fold increase in the tumor content of IF1 was observed in a larger cohort of 37 normal and paired CRC biopsies using “reverse phase protein arrays” (Appendix A). Kaplan–Meier survival analysis revealed a highly significant association between the tumor content of IF1 and disease free survival (DFS) of the patients (Figure 1C). A low content of IF1 in the carcinomas correlated with higher chances of developing metastatic disease (Figure 1C, Table 1).

Univariate Cox regression analysis of seven variables (Table 1) as potential predictors of OS and DFS in colon cancer patients showed significant association of lymph nodes, metastasis, CEA and IF1 expression with OS and DFS of the patients (Table 1). Age also associated with DFS of the patients (Table 1). In multivariate Cox model the results showed that the best predictive factor for OS and DFS was the tumor content of IF1 (Table 1), stressing the relevant role of IF1 as an independent predictor of survival in CRC patients.

### 2.2. Transcriptomic Profiling of HCT116 Cells with Differential Expression of IF1

To investigate the molecular basis of the apparent anti-oncogenic role of IF1 in colon cancer we developed stable HCT116 cell lines overexpressing (IF1) or silencing (shIF1) IF1. The expression of IF1 in the resulting cells was assessed by immunoblotting (Appendix A) and immunofluorescence microscopy (Appendix A) compared to their respective controls. Roughly, a four-fold difference in IF1 expression could be noted between shIF1 and IF1 overexpressing cells.

Analysis of the transcriptome of control (empty vector, EV), shIF1 and IF1 cells revealed that the comparison between silenced and overexpressing cells displayed greater number of differentially expressed genes (Figure 2A and Appendix A). For this reason, subsequent analysis of the transcriptome was performed with the shIF1 vs. IF1 comparison. Bioinformatic analysis showed that from the list of significant genes, 143 were upregulated and 126 were downregulated in shIF1 cells (Figure 2A). A volcano plot that combines the statistical significance with the magnitude of change illustrates the distribution of the IF1-regulated genes emphasizing those that display large magnitude of significant change (Figure 2B). Consistently, the *ATPIF1* gene was found significantly downregulated in shIF1 cells (Figure 2B). For enrichment analysis, we used the Genecodis tool categorizing the genes into Kyoto Encyclopedia of Genes and Genomes (KEGG) pathways. The most affected pathways in shIF1 cells were related to metabolism, pathways in cancer and the cell cycle (Figure 2C).

The set of differentially expressed genes was interrogated with the ingenuity pathways analysis (IPA). This tool is able to predict the activation/repression status of the affected pathways. Unsupervised hierarchical clustering analysis of the 89 genes obtained in IPA confirmed the existence of large differences between shIF1 and IF1 cells (Figure 2D). Differences in the expression of several of these genes were validated by real time PCR confirming the microarray results (Figure 2E). The IPA analysis showed that the majority of activated pathways in shIF1 cells are known to increase the aggressiveness of cancer (Figure 2F). In contrast, the repressed pathways in shIF1 cells were related with cell cycle regulation (Figure 2F), in agreement with the enrichment analysis. Moreover, the analysis of diseases and functions highlighted that the activated pathways in shIF1 cells are related with more aggressive behavior (Figure 2G). Altogether, the results suggest that the overexpression of IF1 in colon cancer cells induces a less invasive phenotype.

### 2.3. Proteomic Analysis of HCT116 Cells with Differential Expression of IF1

Isobaric tags for relative and absolute quantitation (iTRAQ) experiments were performed to identify the major proteomic changes between shIF1 and IF1 cells. A list of 4853 peptides corresponding to 25 protein groups were differentially expressed between shIF1 and IF1 cells as shown in the volcano plot (fold change ≥ 1.5; Figure 3A, see also Appendix A). Hierarchical clustering of the differentially expressed proteins revealed several proteins that were differentially expressed (Figure 3B). The analysis with the Genecodis tool and Panther database showed that proteins upregulated in shIF1 cells are implicated in the regulation of the actin cytoskeleton, cell cycle and migration, anti-apoptosis, tight junction and focal adhesion pathways (Figure 3C). Consistently, the proteins downregulated in shIF1 cells are related to the apoptotic and proteasome degradation pathways (Figure 3C). IPA analysis of the differentially expressed proteins also predicted the activation of cellular movement in shIF1 cells when compared to cells overexpressing IF1 (Figure 3D). Taken together, the results also support at the proteomic level that silencing IF1 is related with an increase in the migration potential of colon cancer cells.

### 2.4. Phenotypic Analysis of HCT116 Cells with Differential Expression of IF1

Overall, the transcriptomic and proteomic data suggested that colon cancer cells with a low expression of IF1 have a higher tumorigenic potential. Indeed, the rates of cellular proliferation showed that shIF1 cells proliferate faster than IF1 cells (Figure 4A,B). In agreement with previous results on the role of IF1 in controlling mitochondrial respiration [11,12,13], cells overexpressing IF1 showed partial inhibition of mitochondrial respiration (Figure 4C). Wound healing assays revealed that shIF1 cells started filling and fully occupied the scratched area earlier than IF1 cells (Figure 4D). Besides, soft-agar colony-formation assays showed that shIF1 cells had more capacity to grow and form colonies in the anchorage-independent assay (Figure 4E). Moreover, cell death assays triggered by treatment of the cells with staurosporine or hydrogen peroxide further revealed that shIF1 cells are more resistant to apoptosis inducing agents than IF1 cells (Figure 4F). Overall, these results support that colon cancer cells with a low content of IF1 have greater potential for tumor growth and metastasis than IF1 overexpressing cells.

### 2.5. In Vivo Tumorigenesis of HCT116 Cells with Differential Expression of IF1 

To assess the tumor growth capacity of the cells in vivo, shIF1 and IF1 cells were subcutaneously injected into the left and right flank of nude mice, respectively (Figure 5A). After implantation, tumor growth and volume were monitored after the administration of luciferin using IVIS Lumina II (Figure 5B,C). The results showed that the differential expression of IF1 had no relevant effect on the in vivo rates of tumor growth of the cells. In contrast, intravenous injection of shIF1 and IF1 cells into nude mice to assess metastatic disease in vivo revealed that shIF1 cells increased the percentage of mice with lung metastases, being the number and growth of metastases bigger (Figure 5D). Altogether, supporting that a low expression of IF1 in colon cancer cells favors metastatic disease.

### 2.6. IF1 Favors Anoikis of Colon Cancer Cells 

Detachment of epithelial cells from the extracellular matrix results in a form of programmed cell death known as anoikis [25]. Importantly, anoikis resistance is a critical mechanism supporting tumor metastasis [25]. Interestingly, shIF1 cells showed more resistance to anoikis than IF1 cells upon cell detachment (Figure 6A). In an attempt to unveil the potential mechanisms that might underpin the anoikis resistance of shIF1 cells we searched for the differential expression of Snai1, a transcription factor involved in EMT, hepatocyte growth factor (HGF) and the activation of PI3K/Akt, mTOR, p38 and ERK1/2 MAPK, NFκB and AMPK signaling pathways, which are positive regulators of anoikis resistance (Appendix A). However, we found no relevant differences in the expression and/or activation of these signaling pathways between shIF1 and IF1 cells (Appendix A). In contrast, cFLIP (FLICE-inhibitory protein), which is a potent inhibitor of receptor-mediated cell death upon cellular detachment from the extracellular matrix [26,27], was significantly overexpressed in shIF1 cells when compared to overexpressing IF1 cells (Figure 6B and Appendix A). Consistently, the tissue content of cFLIP in normal colon was significantly increased when compared to that present in the tumors (Figure 6C and Appendix A), inversely correlating with the expression of IF1 (Figure 1B and Appendix A). Overall, the results support that cFLIP mediated anoikis resistance favors the pro-metastatic phenotype of shIF1 cells.

### 2.7. IF1 Favors Infiltration and Cytotoxicity of NK Cells 

Mitochondria are cellular hubs that signal to the immune system by different molecules that emanate from their activity [5,28]. In this regard, we have recently shown that transgenic mice that overexpress IF1 in colon mitochondria promote the recruitment of anti-inflammatory immune cells in response to inflammation when compared to wild-type mice [29]. Hence, we next investigated the role of natural killer (NK) cells in tumor infiltration and cytotoxicity of tumor spheroids developed from IF1 or shIF1 cells. To this end, colonospheres (Figure 7A) were co-cultured with primary IL-2 and IL-15 pre-activated and expanded CD45-APC stained NK cells at 1:5 ratio and the tumor spheroid integrity, NK infiltration and cytotoxicity was monitored by light and fluorescence microscopy 48 h latter (Figure 7). The infiltration of NK cells in tumor spheroids developed from IF1 overexpressing cells was significantly increased when compared to shIF1 spheroids, as assessed both by immunofluorescence microscopy (Figure 7A) and by flow cytometry (Figure 7B). Eventually, the formation of lytic NK-cell immunological synapses [30] with cancer cells of the spheroids were observed (Figure 7C). Assessment of granzyme B-induced cell death by the quantification of cells showing the activation of the tumor suppressor phospho-p53, indicated a significant increase of dying cells in the IF1 spheroids (Figure 7D). Overall, the results support that a low expression level of IF1 in colon cancer cells favors metastatic disease because the operation of at least two contributing factors: (i) the higher resistance of cancer cells to undergo anoikis upon cell detachment and, (ii) the lower tumor infiltration and cytotoxicity of NK cells observed in these carcinomas. Altogether, the results provide a molecular and functional explanation for the higher incidence of CRC recurrence in patients bearing carcinomas with low expression of IF1 (Figure 1C, Table 1).

## 3. Discussion

The mitochondrial ATP synthase is a critical component regulating energy metabolism, cellular signaling and death, playing the activity and expression of its physiological inhibitor a prominent role [31]. The ATPase Inhibitory Factor 1 is upregulated in different carcinomas [4,12,20,22] triggering metabolic reprograming of the cell by inhibition of mitochondrial respiration and concurrently activating programs aimed at proliferation and survival [4,11]. IF1 is a mitochondrial protein with a very short half-life (2 h) in colon cancer [13] and in stem [6] cells when compared to the half-life of other subunits of the ATP synthase (18 h) [32]. Since the high expression of IF1 observed in CRC is unrelated to changes in the cellular availability of IF1 mRNA [13], we suggest that its accumulation in carcinomas should result from oncogenic transformation of the mechanisms that regulate IF1 mRNA translation and/or the stability of the protein. Evidence for the operation of both mechanisms of posttranscriptional regulation of IF1 expression has already been provided [6,15], thus emphasizing their potential contribution in the pathophysiology linked to IF1 expression.

Not surprisingly, the overexpression of IF1 in hepatocarcinomas [20], bladder [21] and stomach [22] carcinomas and in gliomas [23] contributes, by different mechanisms, to cancer recurrence and progression. Contrariwise, we show that its overexpression in a cohort of carcinomas of the Spanish CRC Epicolon Study [33] identifies patients with less chances of disease recurrence, supporting an anti-metastatic role for IF1. Moreover, we show that IF1 is a significant independent predictor of good prognosis in colon cancer patients. Similarly, the overexpression of IF1 in breast carcinomas also predicts lower chances of disease recurrence [13,24].

To investigate the mechanisms that support the paradoxical anti-oncogenic role of IF1 in colon cancer progression we have developed HCT116-luc colon cancer cells stably expressing low or high IF1 levels. Through a comprehensive transcriptomic, proteomic and in vitro tumorigenic assays we show that colon cancer cells with low IF1 expression have a more pro-oncogenic phenotype with increased migratory and invasive capabilities and increased capacity to evade death than IF1 overexpressing cells. For instance, shIF1 cells revealed deregulated cell cycle, which is a hallmark of cancer cells [34], and predicted the inhibition of the PPAR signaling pathway that is known to inhibit the proliferation of tumor cells [35]. In the same line, a low expression of PPA receptor γ is associated with reduced patients’ survival [36]. On the other hand, p21-activated kinase (PAK), Ephrin receptor, p38 MAPK, ERK/MAPK, HGF, PDGF and renal cell carcinoma signaling, that are known to increase the aggressiveness of cancer, are predicted pathways activated in shIF1 cells at the transcriptomic level, although no evidence for such activations were obtained at the protein level.

Remarkably, although cancer cells reprogram metabolism to divert metabolites into the anabolic pathways that support proliferation [6,7,8,37,38,39], it has been shown that migratory/invasive cancer cells specifically favor mitochondrial respiration and increased ATP production [40,41]. In agreement with this observation, we show that shIF1 cells have a higher respiratory activity of mitochondria than cells overexpressing IF1, also in agreement with similar findings reported in breast cancer cells with low expression of IF1 [24], further stressing the relevant role of mitochondrial bioenergetic function in metastatic disease. Interestingly, although the growth rates of the cells in vitro were statistically different, they had marginal relevance as assessed in vivo, suggesting that the metabolic differences between shIF1 and IF1 cells are unnoticeable for the accretion of tumor mass.

The majority of mortalities caused by colorectal cancer derive from metastatic disease to the liver or lung [42]. When the metastatic potential of the cells was investigated by its ability to colonize the lung, we observed that a low expression of IF1 conferred an increased metastatic potential to shIF1 cells in vivo. Interestingly, shIF1 cells overexpressed cFLIP, a master cell-death regulator that suppresses tumor necrosis factor-α (TNF-α), Fas-L and TNF-related apoptosis-inducing ligand (TRAIL)-induced apoptosis, to prevent the activation of the caspase 8 cascade in the death-inducing signaling complex (DISC) [26,27]. Thus, we suggest that a likely factor that contributes to the increased metastatic potential of shIF1 cells is their increased capacity to evade anoikis.

NK cells represent a first line of defense of the innate immune system because this special type of lymphocytes lyses tumor cells without prior sensitization, thus playing a prominent role in immune surveillance to prevent metastatic disease [43,44]. We show that an additional factor that contributes to the increased metastatic potential of shIF1 cells is the diminished NK cell infiltration and resulting cell-death in shIF1-colonospheres. Importantly, tumor infiltration of NK cells coincides with cytotoxicity and is associated with a better prognosis in colorectal cancer [45]. Cytotoxicity of activated NK cells is exerted by multiple approaches, including the direct lysis of the cancer cell by cytolytic proteins such as perforin and granzymes [46], as we have observed in polarized granules in lytic NK-cell immunological synapses. In addition, shIF1 cells could diminish immune surveillance of NK cells by generating soluble activating receptor ligands, such as cFLIP to block induction of apoptosis by TNF-α, Fas-L/Fas and TRAIL receptors [47]. Interestingly, shIF1 cells reveal significant increased expression of the CXC chemokine receptor 4 (CXCR4), which is involved in the inhibition of the activation and proliferation of NK cells by tumor cells [48,49] (Figure 7E). Moreover, shIF1 cells significantly increased the expression of the transcription factor SMAD3 and of the ecto-5’nucleotidase CD73/NT5E, which is under the control of SMAD3, an enzyme that generates adenosine in the tumor microenvironment leading to the suppression of multiple immune subsets including NK cells [50,51] (Figure 7E). In addition, the mRNA of LDHA (two fold, false discovery rate (FDR) < 0.007) was also significantly increased in shIF1 cells, suggesting that they produce more lactate which might be released to the tumor microenvironment acting also as an immune metabolite that hampers immune surveillance and activation of NK cells [52] (Figure 7E). Altogether, metabolic immune suppression by the tumor microenvironment and enhanced CXCR4 signaling could be involved in lessening immune surveillance of shIF1 carcinomas (Figure 7E).

We should stress the dissimilar behavior reported for IF1 as biomarker in cancer prognosis [31]. In this regard, we suggest that the large differences observed in the tissue-specific expression of IF1 might underlie this behavior [15]. For instance, in the case of liver and stomach, IF1 expression is highly abundant in the normal tissue [13,15] and its overexpression in oncogenesis predicts bad patient prognosis [20,22]. In contrast, the overexpression of IF1 in colon (this study) and breast carcinomas [13,24] is a biomarker of good prognosis. However, at variance with liver and stomach, colon and breast are tissues with low expression of IF1 under normal physiological conditions (this study and [13,24]), suggesting that IF1, in addition to its structural [14] and inhibitory functions on the ATP synthase [19], has additional tissue-specific functional roles [15,31].

Altogether, the results indicate that an elevated expression of IF1 in CRC predicts a favorable prognosis for colon cancer patients. Indeed, the overexpression of IF1 in CRC represents a great advantage over carcinomas with low expression of IF1 because in the latter case metastatic disease is favored by the ability of the cells to escape anoikis and immune surveillance by NK cells.

## 4. Materials and Methods

### 4.1. Patient Specimens and IF1 Quantification

A collection of frozen tissue sections obtained from surgical specimens of untreated cancer patients with primary colorectal adenocarcinomas, enrolled in the incident Spanish CRC Epicolon Study and prospectively followed during 5 years were included in this study [33]. Patients’ medical records were reviewed and identifiers coded to protect patient confidentiality. Normal and tumor samples, supplied by the Banco de Tejidos y Tumores, Department of Pathology, Hospital Meixoeiro, Vigo, Spain, were provided with the understanding and written consent from the patients. The study methodologies conformed to the standards set by the Declaration of Helsinki and were approved by the Institutional Review Board “CEIC de Galicia: 2010/039”. The clinicopathological characteristics and IF1 content in paired normal and tumor biopsies of the CRC patients studied are provided in Appendix A. The content of IF1 in normal (2.2 ± 0.1 pg/ng protein) and tumor (3.6 ± 0.4 pg/ng protein) biopsies of this cohort of CRC were determined by reverse phase protein arrays using recombinant hIF1 [13].

### 4.2. Generation of Cell Lines, Cell Cultures and Spheroids

The human HCT116 and HCT116-luc (Caliper Life Sciences, Inc., Hopkinton, MA, USA) colon cancer cells were cultured in McCoy’s 5A media. For the IF1 overexpression approach, the pCDH-CMV-MCS-EF1-RFP + PURO cDNA Cloning and Expression Vector (SBI#CD616B-2; SBI System Biosciences, Palo Alto, CA, USA) was used to establish cells overexpressing the human ATPase IF1 [24]. For the IF1 silencing approach, we used the pEco-Lenti-U6 shRNA-(RFP-puro) (GenTarget Inc, San Diego, CA, USA. # LTSH-U6-RP) silencer plasmid encoding the IF1 shRNA3 (Invitrogen, Thermo Fisher Scientific, Waltham, MA, USA) driven by the U6 promoter [24]. Stable cell lines were generated using lentivirus [24]. Viral particles were produced in HEK293T cells. After transfection, stable transfectants were selected by adding 6 μg/mL puromycin (Invitrogen) to the growth medium.

For spheroid formation, 3 × 103 IF1 or shIF1 HCT116 cells were seeded for each spheroid. After three days, the spheroids were transferred to non-adherent Nunclon Sphera™ 96F-well plate (ThermoFisher Scientific). The addition of CD45-APC stained NK cells was done one day after.

### 4.3. NK Cell Isolation and Labeling

Primary NK cells were isolated from buffy coats of healthy donors provided by Centro de Transfusión de la Comunidad de Madrid (Spain) obtained according to the Principles of the Declaration of Helsinki. Peripheral blood mononuclear cells (PBMCs) were isolated by density centrifugation on Lymphoprep (Stemcell Technologies, Vancouver, BC, Canada) and Percoll (GE Healthcare, Chicago, IL, USA). PBMCs from the 1.068 density layer were cultured for 1 week in RPMI 1640 (Lonza, Basel, Switzerland) with 10% FBS (Gibco, Thermo Fisher Scientific, Waltham, MA, USA), 5% human AB serum (Sigma-Aldrich, St. Louis, MO, USA), 10 mM HEPES, penicillin/streptomycin (10 U/mL) and 2 mM L-glutamine (Gibco, Thermo Fisher Scientific, Waltham, MA, USA), 50 IU/mL of recombinant human rhIL-2 (Peprotech, Rocky Hill, NJ, USA) and 20 IU/mL of rhIL-15 (NISBC, Blanche Lane, UK). For NK cell purification, cultures were depleted of T, B and myeloid cells by magnetic sorting using CD4-PE (Beckman Coulter, Brea, CA, USA), CD3-PE, CD19-PE, CD13-PE mAbs (BD Biosciences, San Jose, CA, USA) and anti-PE microbeads (Miltenyi Biotech, Bergisch Gladbach, Germany). Negative cells containing NK cells (>80% purity) were isolated by LS columns (Miltenyi Biotech, Bergisch Gladbach, Germany). For spheroid co-cultures, NK cells were labeled with APC-coupled anti-human CD45 or mouse IgG1 control antibodies (BD Biosciences).

### 4.4. Cellular Lysis and Western Blotting

Cell lysis was performed with RLN-T buffer (RLN buffer plus 0.5% Triton X-100 and the complete protease inhibitors cocktail EDTA-free; Roche, Basel, Switzerland) at 20 × 106 cells/mL for 5 min on ice and cellular extracts prepared [53]. See Appendix A for antibodies used in western blotting.

### 4.5. Gene Array Hybridization

Total RNA was extracted using Trizol (Invitrogen) followed by the Qiagen RNeasy (Qiagen, Hilden, Germany). Total RNA (200 ng) was amplified using the One Color Low Input Quick Amp Labeling Kit (Agilent Technologies) and purified with the RNeasy Mini Kit (Qiagen). Preparation of probes and hybridization was performed as described in One-Color Microarray Based Gene Expression Analysis Manual Ver. 6.5, Agilent Technologies as recently described in detail [24]. Slides were Sure Print G3 Agilent 8 × 60K Human (G4852A-028004). Images were captured with an Agilent Microarray Scanner and spots quantified using Feature Extraction Software (Agilent Technologies, Santa Clara, CA, USA). Hybridizations and statistical analysis were performed by the Genomics Facility at Centro Nacional de Biotecnologia (Madrid, Spain).

### 4.6. Gene Data Analysis

Genes with a fold change ≥2 or ≤−2 between control, overexpressing and silenced IF1 cells were selected. A corrected *p*-value ≤ 0.01 was considered statistically significant. Genecodis3 tool was used to perform gene set enrichment analysis using KEGG and Panther pathways enrichments. Gene set enrichment by Ingenuity Pathway Analysis (Qiagen) was also used to determine the specific canonical signaling pathways and related diseases and functions affected. A Fisher’s exact right-tailed test identified significantly enriched pathways, and a z score was provided.

### 4.7. Quantification of mRNA

Reverse transcription reactions were performed using 1 μg of total RNA and the High Capacity Reverse Transcription Kit (Applied Biosystems; Thermo Fisher Scientific, Inc.). Real-time polymerase chain reaction (PCR) was performed with an ABI PRISM 7900HT SDS (Applied Biosystem, Waltham, MA, USA) and Power SYBR Green PCR Master Mix (Applied Biosystems). See Appendix A for primers used.

### 4.8. Proteomic Analysis and ITRAQ Labeling

Disulfide bonds from cysteinyl residues in proteins were reduced with 10 mM DTT and the thiol groups alkylated with 50 mM iodoacetamide. Proteins were digested in situ with sequencing grade trypsin (Promega). Digestion was stopped by the addition of 1% TFA. Whole supernatants were dried down and then desalted onto OMIX Pipette tips C18 (Agilent Technologies, Santa Clara, CA, USA). The peptides from tryptic digests (100 µg) were labeled using chemicals from the iTRAQ reagent 8plex Multi-plex kit (reagents 113, 114, 115, 116, 117, 118, 119 and 121; Applied Biosystems). Labeling was stopped by the addition of 0.1% formic acid. Supernatants were dried down and the eight samples were mixed to obtain the “8plex-labeled mixture”. The mixture was desalted onto OMIX Pipette tips C18 until mass spectrometric analysis.

### 4.9. Reverse Phase-Liquid Chromatography RP-LC-MS/MS and Data Analysis

The desalted 8plex-labeled mixture was dried, resuspended in 10 µL of 0.1% formic acid and analyzed by Reverse Phase-Liquid Chromatography coupled to tandem Mass Spectrometry (RP-LC-MS/MS) in an Easy-nLC II system coupled to an ion trap LTQ-Orbitrap-Velos-Pro hybrid mass spectrometer (Thermo Scientific) as recently described in detail [32]. The instrument method consisted of a data-dependent top-20 experiment with an Orbitrap MS1 scan at a resolution (m/Δm) of 30,000 followed by either twenty high energy collision dissociation (HCD) MS/MS mass-analyzed in the Orbitrap at 7500 (Δm/m) resolution. MS2 experiments were performed using HCD to generate high resolution and high mass accuracy MS2 spectra. The minimum MS signal for triggering MS/MS was set to 500. The lock mass option was enabled for both MS and MS/MS mode and the polydimethylcyclosiloxane ions (protonated (Si(CH3)2O))6; m/z 445.120025) were used for internal recalibration of the mass spectra. Peptides were detected in survey scans from 400 to 1600 amu (1 μscan) using an isolation width of 2 u (in mass-to-charge ratio units), normalized collision energy of 40% for HCD fragmentation, and dynamic exclusion applied during 30 seconds periods. Precursors of unknown or +1 charge state were rejected.

Protein identification and quantification from raw data was carried out using the SEQUEST algorithm (Proteome Discoverer 1.4, Thermo Scientific) and PEAKS 8 (Bioinformatics Solutions, Waterloo, ON, Canada). Database search was performed against uniprot-Homo.fasta with the constrains described in detail [32] and iTRAQ reagent labeling at the N-terminus and lysine residues. Search against decoy database using false discovery rate (FDR) < 0.01. All proteins were identified with at least two high confidence peptides.

### 4.10. Immunohistochemistry

Formalin-fixed normal and tumor specimens of the colon were immunostained using the monoclonal anti-IF1 (1:200) antibody [12]. Sections were counterstained with hematoxylin.

### 4.11. Cellular Proliferation, Cell Death Assays and Cellular Respiration

Cellular proliferation was determined by protein concentration after 24 h, 48 h, 72 h and 96 h of culture and by the incorporation of 5-ethynyl-20deoxy-uridine (EdU) into cellular DNA using the Click-iT EdU Flow Cytometry Assay Kit (Molecular Probes, Thermo Fisher) [11]. For cell death assays, 50,000 cells/well were seeded and treated with staurosporine (STS) or hydrogen peroxide (H2O2). Cell death was determined by flow cytometry after staining with Annexin V (ApoScreen FITC; SouthernBiotech, USA and Canada) [11]. Oxygen consumption rates were determined in an XF24 Extracellular Flux Analyzer (Agilent Technologies) using 10 mM glucose, 1 mM pyruvate and 2 mM glutamine. Cells were seeded in the microplates, and incubated at 37 °C and 5% CO_2_ for 24 h. The final concentration and order of injected substances was 6 μM oligomycin, 0.75 mM DNP (2,4-dinitrophenol), 1 μM rotenone and 1 μM antimycin.

### 4.12. Soft-Agar, Wound Healing, Invasion and Anoikis Assays

For soft-agar assays, 2000 cells were used and layered onto complete growth media plus 0.7% of agar as recently described in detail [24]. Viable colony numbers were stained and counted using ImageJ software. For wound healing assays [24], confluent cell monolayers were mechanically disrupted to produce a uniform scratch and photographed every 30 minutes. Gap distances were analyzed with Image J software. Corning Biocoat Matrigel Invasion Chambers (8.0 µm pore size) were used to quantify the cellular invasive capacity. A total of 2 × 104 cells were seeded in 1% FBS and chemoattraction perform during 72 h in 20% FBS. Anoikis resistance was determined using Cytoselect 96-well Anoikis Assay (Cell Biolabs, San Diego, CA, USA). 12,000 cells were seeded into each well, and 24 hours later cell viability and cell death were measured by Calcein Am and EthD-1 fluorometric detection, respectively.

### 4.13. In Vivo Assays

Mice experiments were carried out in accordance with EU Directive 2010/63/EU for animal experiments and the methodologies approved by the Ethical Committee of the Universidad Autónoma de Madrid (CEI-52-961). Approximately, 4 × 10^6^ HCT116-luc shIF1 cells were injected subcutaneously into the left flank of 6-week-old nude mice (Envigo, Huntingdon, Cambridgeshire, UK). The same number of HCT116-luc IF1 overexpressing cells was injected into the right flank of the same animals. Tumor growth was followed every week after implantation of the cells by the intraperitoneal injection of 150 mg/kg of body weight of D-luciferin (Promega, Madison, WI, USA) utilizing the IVIS Lumina II equipment (Caliper Life Sciences, Hopkinton, MA, USA). Tumor size was determined using a standard caliper [53]. Following the ethical criteria established by our Institutional Review Board the animals were killed when the tumor volume reached 2000 mm^3^. To study metastasis in vivo, female nude mice were inoculated with 8 × 10^6^ HCT-luc IF1 overexpressing or HCT-luc IF1 silenced cells (seven mice per group) through the tail vain. Metastatic disease was monitored by in vivo imaging of the anesthetized mice after the i.p. administration of luciferin using the IVIS Lumina II imaging system. Animals were followed during a maximum of 4 weeks and euthanized under CO2 stream followed by cervical dislocation.

### 4.14. Flow Cytometry Analyses of Spheroid Invasion

IF1 and shIF1 HCT116 cells were stained with CellTrace™ Violet (CTV) (ThermoFisher, C34557) in serum-free DMEM media (1:1,000) for 20 minutes at room temperature. CountBright™ Absolute Counting Beads (Cat. No. C36950, ThermoFisher Scientific) were used to quantify the number of invading NK cells (CD 45+) by flow cytometry. For each analysis, 10,000 events were recorded.

### 4.15. In-Vivo Confocal Microscopy of NK cell Invasion and Cytotoxicity

In-vivo analysis of the interaction between NK cells and colonospheres was performed on Nuclon Sphera 96F bottom plate (Cat. No. 174927, ThermoFisher Scientific) in RPMI medium supplemented with IL-2 and IL-15. Pictures were taken in an inverted Axiovert200 (Zeiss, Oberkochen, Germany) microscope with a scientific complementary metal-oxide semiconductor (sCMOS) monochrome camera and 10× objective, for 48 h at 37 °C and 0.5% CO_2_. The anti-CD45 APC conjugated antibody (BD Bioscience) was used at a 1:10 dilution to stain NK cells. Immunofluorescence microscopy of 4% PFA-fixed colonospheres was used to analyze the infiltration of NK cells. The anti-Phospho-p53 (Ser15) antibody (Cell Signaling) was used at a 1:100 dilution to assess the rate of cell death in colonospheres. Nuclei were counterstained with DAPI (diamidino-2-fenilindol) 1:5000. Cellular fluorescence was analyzed by confocal microscopy in a Nikon A1R + microscope. The area and cell number were quantified by ImageJ 1.51w software.

### 4.16. Statistical Analysis

The results shown are the means ± SEM. Statistical analysis were performed by Student’s *t*-test, Mann–Whitney U test and/or Kruskal Wallis as appropriate. Tests were calculated using the SPSS 13.0 software package (IBM, Chicago, IL, USA). Survival curves were derived from Kaplan–Meier estimates and compared by log-rank test. The mean value of IF1 expression in normal samples was used as cut-off in Kaplan–Meier. The hazard ratio (HR) with 95% confidence interval (CI) was calculated using univariate and multivariate Cox proportional hazards regression analysis. Statistical tests were two-tailed at the 5% level of significance.

## 5. Conclusions

In summary, we demonstrated that the IF1 content in colorectal carcinomas was an independent predictor of prognosis for CRC patients. In contrast to the pro-oncogenic role of IF1 in other carcinomas, the overexpression of IF1 in CRC is a biomarker of good prognosis. Molecular and functional analyses of colon cancer cells with differential expression of IF1 indicate that cells overexpressing IF1 have lower mitochondrial respiration and a less aggressive phenotype because they are more vulnerable to death upon cell detachment. In vivo assays confirmed the higher metastatic potential of the cells with higher respiratory activity as a result of a low mitochondrial content of IF1. Moreover, infiltration and cytotoxicity of NK cells in tumor spheroids confirmed that carcinomas with high IF1 content were more vulnerable to immune surveillance. Taken together, these data indicated that IF1 was a biomarker and prognostic factor of CRC and provided the molecular bases for its tumor suppressor function in colon cancer.

## Figures and Tables

**Figure 1 cancers-12-00022-f001:**
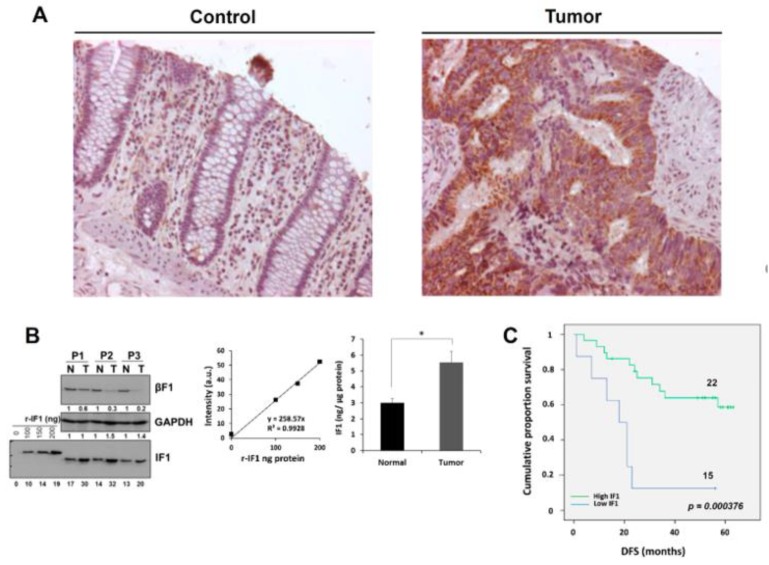
Expression of IF1 in human colon carcinomas. (**A**) Representative staining of IF1 expression in normal and tumor tissue of the colon. Magnification 25x. (**B**) Western blots show IF1 and β-F1-ATPase (β-F1) in paired normal (N) and tumor (T) biopsies derived from three representative patients and increasing amounts of the recombinant IF1 (r-IF1) protein (0–200 ng). Linear correlations between the fluorescence intensity (arbitrary units) of the spots and the amount of recombinant protein. The histogram shows significant differences in the expression of IF1 between the two groups of samples. Protein concentration was calculated according to the fluorescence intensity obtained in r-IF1 plot. The results shown are mean ± S.E.M; *, *p* < 0.05 when compared to its respective control. (**C**) Kaplan–Meier curves for disease-free survival probability for the cohort of 37 colon cancer patients stratified by the tumor expression level of IF1. The log-rank test *p*-value (*p* < 0.0004) is shown.

**Figure 2 cancers-12-00022-f002:**
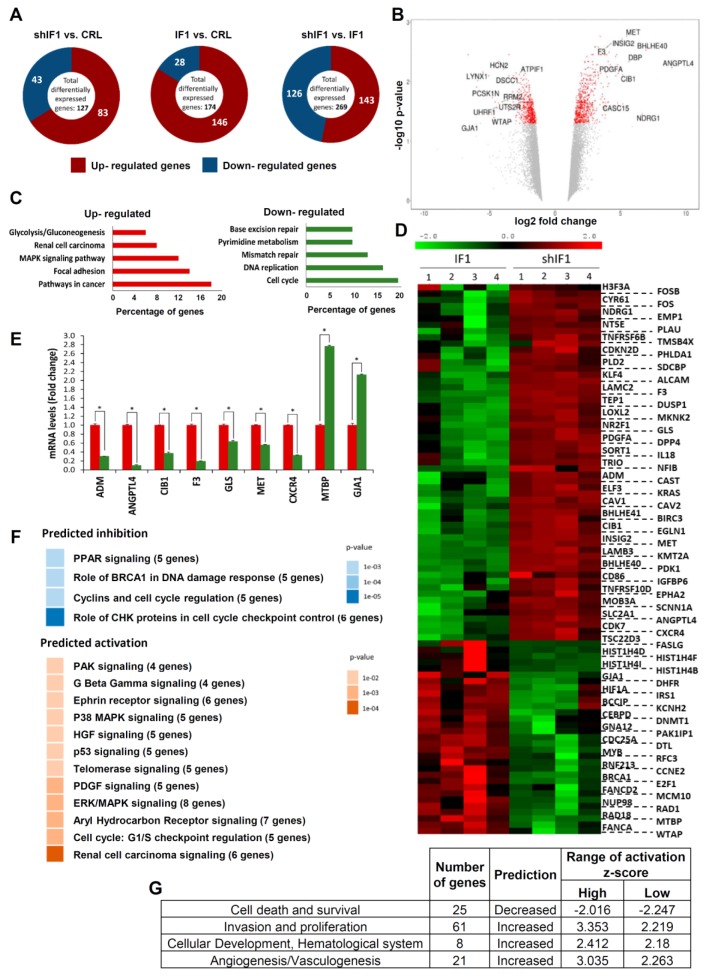
Transcriptome of colon cancer IF1-overexpressing and IF1-silenced cells. (**A**) Representation of the total number of significantly affected genes in the comparisons between four different preparations (1–4) of control, silenced and overexpressing IF1 cells using Agilent 8 × 60K Human arrays. (**B**) Volcano plot with some relevant genes indicated. X axis represents the expression fold change of the affected genes and the Y axis represents –log10 of the false discovery rate (FDR) values. (**C**) Gene enrichment analysis, showing the information related to KEGG. (**D**) Hierarchical clustering analysis using differentially expressed genes implicated in IPA pathways. Four different samples of each cell type were included in the arrays. (**E**) Quantitative reverse transcription PCR validation of up- and down-regulated genes in the microarray analysis in shIF1 (red bars) and IF1 (green bars) cells. *, *p* ≤ 0.05 by Student’s *t* test. (**F**,**G**) Pathways (**F**) and diseases and functions (**G**) affected by silenced IF1 cells as reveal by the IPA ingenuity tool. Z-score indicates the overall predicted activation/inhibition state of the function.

**Figure 3 cancers-12-00022-f003:**
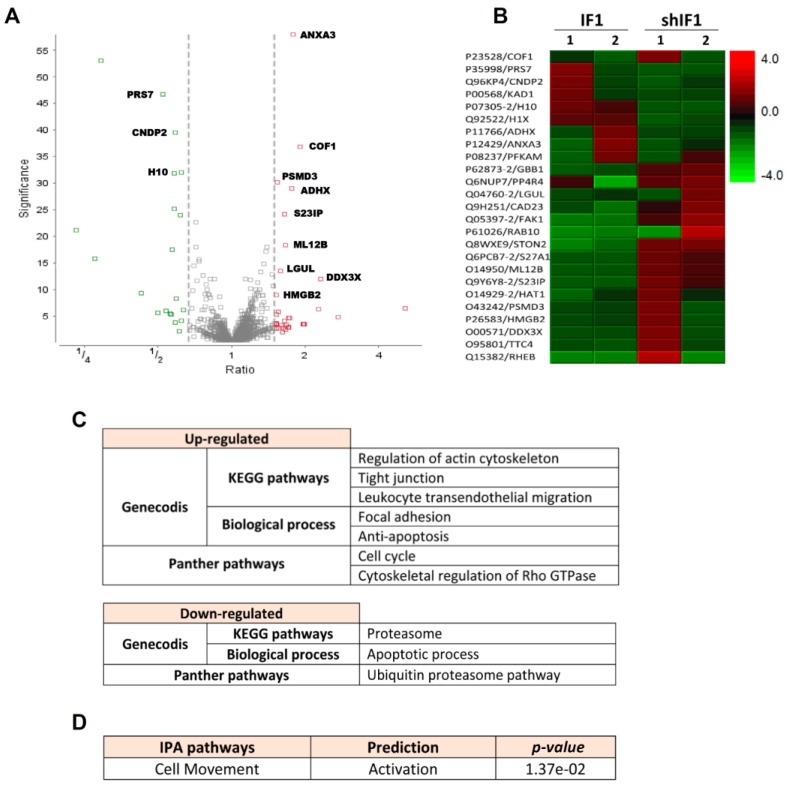
iTRAQ analysis of two different (1–2) preparations of IF1 and shIF1 cells. (**A**) Volcano plot of the proteins quantified in iTRAQ analysis. The plot shows the ratio (shIF1/IF1) and significance on the X and Y axes, respectively. (**B**) Hierarchical clustering analysis based on significant differences in protein expression level between IF1-silenced and overexpressing cells. (**C**) Pathways affected in shIF1 cells analyzed by Genecodis and Panther. (**D**) Activated pathway in shIF1 cells analyzed by IPA.

**Figure 4 cancers-12-00022-f004:**
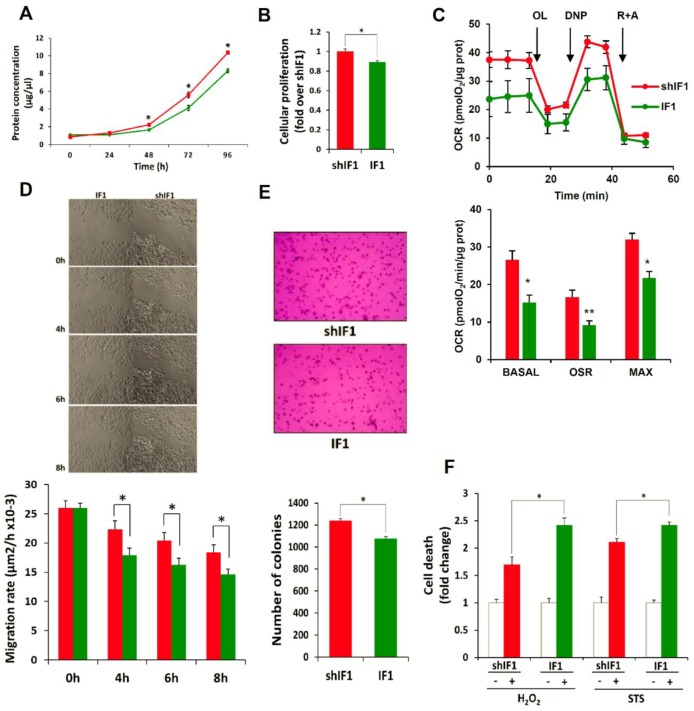
IF1-silenced colon cancer cells have more aggressive phenotype. Cellular proliferation in shIF1 (red lines and bars) and IF1 (green lines and bars) was assessed by protein concentration (*n* = 3; **A**) and by the incorporation of EdU (*n* = 6; **B**) into cellular DNA. (**C**) Rates of basal, oligomycin sensitive and maximum respiration in shIF1 and IF1 cells determined in the X24 Seahorse Flux Analyzer (*n* = 3). OL, oligomycin; DNP, dinitrophenol; R + A, rotenone + antimycin A. (**D**) Time frames of the wound-healing assay and its quantification (*n* = 3). Magnification 4x. (**E**) Anchorage-independent growth in soft-agar and its quantification (*n* = 4). Magnification 10x. (**F**) Cell death determination by flow cytometry using Annexin V staining after overnight treatment with 0.25 µM staurosporine (STS) and 60 µM hydrogen peroxide (H_2_O_2_) (*n* = 5). (**A**–**F**) Results shown are means ± S.E.M; *, *p* ≤ 0.05 by Student’s *t* test.

**Figure 5 cancers-12-00022-f005:**
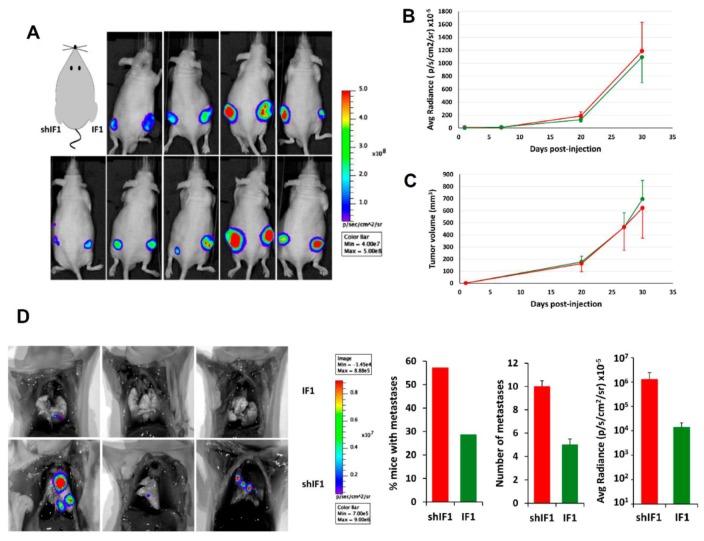
In vivo tumorigenesis and metastasis assays. IF1-silenced and overexpressing HCT116 cells were injected (**A**–**C**) into the left and right flank of sixteen mice, respectively or (**D**) into the tail vein of seven mice per condition. (**A**) Shows bioluminescence imaging of luciferase-positive HCT116 tumors 4 weeks after injection. The graphs show the bioluminescence (**B**) and tumor volume (**C**) of shIF1- and IF1-tumors. (**D**) Bioluminescence imaging of lung metastasis in mice injected with IF1 and shIF1 cells. The histograms show the percentage of mice with metastases, the number of metastases and its bioluminescence. Results shown are means ± S.E.M. IF1, IF1-overexpressing cells are represented by green bars and shIF1, IF1-silencing cells by red bars.

**Figure 6 cancers-12-00022-f006:**
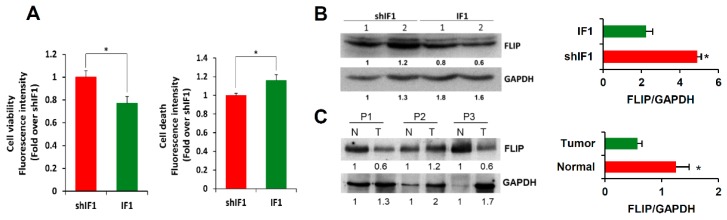
IF1-silenced cells are more resistant to anoikis. (**A**) Cell viability and cell death induced by anoikis in IF1-overexpressing and shIF1 cells (*n* = 4). (**B**) Western blot analysis of the expression of FLIP in two replicates of IF1 silenced and overexpressing cells (*n* = 4). (**C**) Western blot analysis of the expression of FLIP in paired normal (N) and tumor (T) biopsies derived from three CRC patients. Results shown are means ± S.E.M; *, *p* ≤ 0.05 by Student’s *t* test.

**Figure 7 cancers-12-00022-f007:**
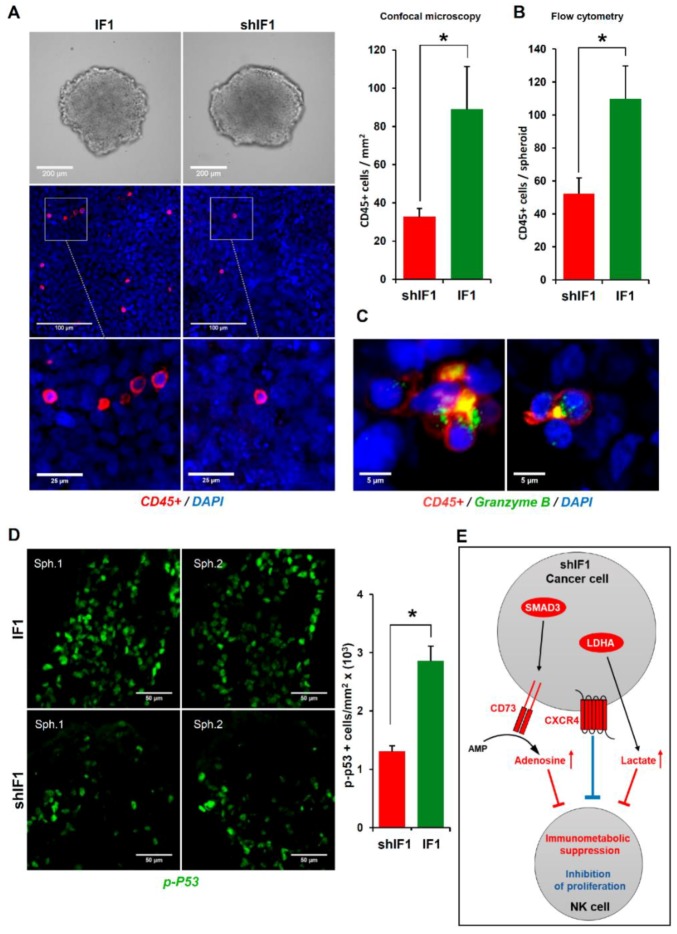
IF1-silenced colon cancer spheroids were more resistant to NK cells invasion and cytotoxicity. (**A**) Bright field microscopy of IF1-overexpressing and IF1-silenced spheroids without NK cells (upper panels) and confocal microscopy images (middle panels) from IF1-overexpressing and IF1-silenced spheroids co-cultured for 48 h with CD45-APC stained NK cells. Lower panels are a zoom of the indicated area in middle panels. Nuclei: blue, DAPI staining; CD45+ cells: red. Histograms show the quantification of CD45+ cells/mm2 by immunofluorescence microscopy (*n* = 5) (**A**) and CD45+ cells/spheroid by flow cytometry (**B**). For flow cytometry, results shown are means of eight IF1-overexpressing or IF1-silenced spheroids ± S.E.M. (**C**) Confocal microscopy images of lytic NK-immunological synapses. Nuclei: blue; CD45+ cells: red; Granzyme B: green. (**D**) Colon cancer cytotoxicity by NK cells was assessed by quantification of phospho-p53 staining (green) by confocal microscopy. Two representative preparations (Sph1 and Sph2) are shown for IF1-overexpressing and IF1-silenced spheroids co-cultured with NK cells for 48 h. Histogram shows the quantification of p-p53+ cells/mm2 (*n* = 4). *, *p* ≤ 0.05 by Student’s *t* test. IF1, IF1-overexpressing cells are represented by green bars and shIF1, IF1-silenced cells by red bar. (**E**) Hypothetical scheme based on transcriptomic data (Appendix A) showing the increased expression of the CXC chemokine receptor 4 (CXCR4, three fold, FDR < 0.005), transcription factor SMAD3 (two fold, FDR < 0.04), ecto- 5’nucleotidase CD73/NT5E (three fold, FDR < 0.005) and LDHA (two fold, FDR < 0.007) in shIF1 cells that might contribute to lessening immune surveillance. Adenosine and lactate in the microenvironment contribute to immunometabolic suppression (red) whereas CXCR4 signaling inhibits activation and proliferation of NK cells (blue).

**Table 1 cancers-12-00022-t001:** Univariate and multivariate Cox regression analysis for overall survival and disease-free survival in colorectal cancer patients.

**Univariate Analysis**	**Overall Survival**		**Disease-Free Survival**	
**Variable**	**HR (95% CI)**	***p*-Value**	**HR (95% CI)**	***p*-Value**
**Age (year)**				
≤70	1 (Reference)		1 (Reference)	
>70	3.01 (0.97–9.27)	0.055	3.67 (1.20–11.23)	0.022
**Gender**				
Female	1 (Reference)		1 (Reference)	
Male	0.59 (0.22–1.53)	0.280	0.58 (0.22–1.47)	0.252
**Grade**				
I	1 (Reference)		1 (Reference)	
II	2.94 (0.38–22.62)	0.300	3.56 (0.46–27.27)	0.221
III	7.094 (0.78–63.99)	0.081	8.62 (0.95–78.21)	0.055
**Lymph nodes**				
No	1 (Reference)		1 (Reference)	
Yes	4.62 (1.49–14.30)	0.008	3.61 (1.28–10.23)	0.015
**CEA**				
≤5 µg/L	1 (Reference)		1 (Reference)	
>5 µg/L	2.79 (1.06–7.36)	0.038	2.64 (1.02–6.86)	0.045
**Metastasis**				
No	1 (Reference)		1 (Reference)	
Yes	4.34 (1.57–11.94)	0.004	3.87 (1.42–10.52)	0.008
**IF1**				
Low	1 (Reference)		1 (Reference)	
High	0.176 (0.06–0.49)	0.001	0.19 (0.07–0.52)	0.001
**Multivariate Analysis**	**Overall Survival**		**Disease-Free Survival**	
**Variable**	**HR (95% CI)**	***p*-value**	**HR (95% CI)**	***p*-value**
**Age (year)**				
≤70	-------	-------	1 (Reference)	
>70	-------	-------	2.50 (0.76–8.21)	0.131
**Lymph nodes**				
No	1 (Reference)		1 (Reference)	
Yes	3.18 (0.95–10.61)	0.059	2.25 (0.71–7.13)	0.165
**CEA**				
≤5 µg/L	1 (Reference)		1 (Reference)	
>5 µg/L	0.96 (0.17–5.20)	0.960	1.31 (0.21–7.98)	0.767
**Metastasis**				
No	1 (Reference)		1 (Reference)	
Yes	3.23 (0.55–18.77)	0.191	1.94 (0.31–12.08)	0.476
**IF1**				
Low	1 (Reference)		1 (Reference)	
High	0.250 (0.08–0.76)	0.015	0.32 (0.10–0.99)	0.049

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
