# Peer review of "Overexpression of Mitochondrial IF1 Prevents Metastatic Disease of Colorectal Cancer by Enhancing Anoikis and Tumor Infiltration of NK Cells"

_cancers, 2019, doi:10.3390/cancers12010022_

Round 1

Reviewer 1 Report

In this manuscript Gonzalez-Llorente L., et al. described results of investigation conducted with colorectal carcinoma cells expressing different level of ATPase inhibitory factor 1 (IF1). The authors revealed that in contrast to other carcinomas (with pro-oncogenic effect of IF1) IF1 overexpression in colon tumor cells was associated with the tumor suppression. The study demonstrated that the mechanisms of anticancer effect of IF1 overexpression was connected with increased intensity of apoptosis (anoikis), low metastatic potential, and with heightened infiltration and cytotoxicity of NK cells in such type of tumors. The obtained results indicate that IF1 is a biomarker and prognostic factor for patients with colorectal carcinoma.

Manuscript is well written and the conclusions are convincingly supported by experimental results. one suggestion will improve the overall quality of the manuscript:

There was no significant difference between in vivo tumor growth after shIF1 and IF1 cell injection into nude mice (Fig. 5, A-C). Providing an explanation of these results will be useful for the manuscript.

Author Response

Manuscript is well written and the conclusions are convincingly supported by experimental results. one suggestion will improve the overall quality of the manuscript:

There was no significant difference between in vivo tumor growth after shIF1 and IF1 cell injection into nude mice (Fig. 5, A-C). Providing an explanation of these results will be useful for the manuscript.

We thank the reviewer for supporting our work.

In answer to the timely question raised by the reviewer we could say that despite the growth rates of the cells in vitro were statistically different, the differences noted were low or of marginal relevance (see comment of R#3). This situation might explain per se why we observed no growth differences of the tumors in vivo. Apparently, the observation suggests that the metabolic differences noted between shIF1 and IF1 cells are not relevant for the accretion of tumor mass. The differences are relevant in vivo when metastasis is taken into consideration, when the reliance of the cells on OXPHOS and the expression of cFLIP provides a clear advantage for colonizing the lung to shIF1 cells. As suggested by the reviewer, a comment in this regard has been included in the revised version of the paper (See lines 306-308).

Reviewer 2 Report

The manuscript by Liorente et al, on “Overexpression of mitochondrial IF1 prevents metastatic disease of colorectal cancer by enhancing anoikis and tumor infiltration of NK cells” investigated the functional role of mitochondrial IF1 in CRC patients. The study is well designed and the experiments were sequentially performed well. The data strongly support the conclusion made and the methods used are well documented. Overall, these interesting study reveals the important anti-metastatic role of IF1 in CRC patients.

Minor corrections

Fig.1D, graphs are switched with the immunoblotting.

Fig.1E, in the graph, comparison of shIF1 vs shC is reliable instead of comparing with the control sample. The text explaining the immunofluorescence could be improved.

Fig.4D, in the wound healing assay, 6h data is missing in the representative figure, with 0h missing in the graph.

Figures 1-6, indicate the ‘n’ used in these experiments in figure legends.

Author Response

Minor corrections

We thank the reviewer for the support of our work.

Fig.1D, graphs are switched with the immunoblotting.

We thank the reviewer for the observation and we apologize for the error. In the revised new Figure 1D, we have changed the position of the graphs.

Fig.1E, in the graph, comparison of shIF1 vs shC is reliable instead of comparing with the control sample. The text explaining the immunofluorescence could be improved.

We agree with the reviewer’s comment. In the revised new Figure 1E, we have incorporated the histogram to illustrate the comparison of shIF1 vs shC cells. The corresponding figure legend has been modified accordingly. In addition, we have incorporated a sentence in the text (see line 114-115) stressing that roughly a four-fold difference in IF1 expression could be noted between shIF1 and IF1 overexpressing cells.

Fig.4D, in the wound healing assay, 6h data is missing in the representative figure, with 0h missing in the graph.

We thank the reviewer for the observation. In the revised new Figure 4D in the wound healing assay, we have now included the missing 6h picture and the value of 0h point in the graph.

Figures 1-6, indicate the ‘n’ used in these experiments in figure legends.

In the new Figure Legends, we now indicate the corresponding “n” used in the experiments.

Reviewer 3 Report

González-Llorente et al. describes the role of mitochondrial IF1 in CRC and shows that overexpression of IF1 prevents metastatic disease of colorectal cancer. The authors then show that IF1 promote anoikis and tumor infiltration of NK cells to prevent metastatic CRC.

There are several concerns which should be addressed,

In line 45, the author states that IF1 is negligibly expressed in normal human colon but show expression levels of IF1 in panel 1B. Can the authors explain this?

The authors should include the text of entire Fig 1 in one paragraph and not three different paragraphs or should split the figures accordingly as it is difficult to follow in present form

The author states in line 97 that low content of IF1 in carcinomas correlated with higher chances of developing metastatic disease and gave reference of Fig 1C. This is not correct it should be table 1. Even in the figure legend 1, line 91 states that the log-rank test p-value is shown and gives reference of Fig 1D, 1E. This is also incorrect as it relate to knockdown and overexpression of IF1.

The quality of Fig 2 is poor. The authors should work on it especially the heatmap where it is difficult to read. The authors could highlight only important genes.

Line 131 states the IPA analysis showed that the majority of activated pathways in shIF1 cells were related to cancer??

Although the authors state the reason for comparison between shIF1 and IF1, the real comparison should be with their respective controls and not between knockdown and IF1. 

The cell growth assays in Figure 4A shows that shIF1 grew more than IF1 although significant; the quantification in 4B is marginal. Moreover if this is true, there is no difference between shIF1 and IF1 in vivo in xenograft models.

The authors found that c-FLIP mediated anoikis resistance in shIF1 cells. If this is the case then authors should test this in N and T samples of CRC patients and show that c-FLIP levels are reduced in tumors when they express IF-1 and show that knockout of c-FLIP in shIF1 cells lose their metastatic potential.

The authors state that overexpression of IF1 in colon mitochondria promote the recruitment of anti-inflammatory immune cells in response to inflammation and refer to [29]. In that manuscript the authors describe about macrophages and not NK cells. So what was the rationale of looking at NK cells and not others like neutrophils or macrophages?  If NK cells play important role in tumor infiltration in spheroid models, the authors also show this relevance in CRC patients who have increased IF1 levels

Minor points

Line 17 in abstract - less chances of disease recurrence, being IF1 (change to IF1 being) an independent predictor of survival

The reference 1 is old and should be updated with 2019 statistics

In panel 1C in the figure legend, the author states Kaplan-Meier curves for disease-free survival probability for the cohort of 38 colon cancer patients but in Fig they show only for 37 patients

Fig 1D the quantification is switched for knockdown and overexpression of IF1

The quantification for Fig 1E should include four groups of Ctrl, IF1, shC and shIF1

Author Response

There are several concerns which should be addressed,

 We thank the reviewer for the evaluation and improvement of our work.

In line 45, the author states that IF1 is negligibly expressed in normal human colon but show expression levels of IF1 in panel 1B. Can the authors explain this?

In agreement with the reviewer’s comment, it should read that “IF1 is expressed at low levels in normal human colon”. The text has been modified accordingly. The sharper differences noted in IF1 expression between immunohistochemistry (Fig 1A) and western blotting (Fig 1B) could arise by the preservation and/or unmasking of epitopes during tissue fixation and/or retrieval of antibody epitopes in immunohistochemical procedures.

The authors should include the text of entire Fig 1 in one paragraph and not three different paragraphs or should split the figures accordingly as it is difficult to follow in present form

We found it really difficult to follow the reviewer’s request. However, to improve presentation we have shifted the beginning of section “2.2 IF1 is an independent….” to the beginning of page 3, before presenting Fig. 1.

The author states in line 97 that low content of IF1 in carcinomas correlated with higher chances of developing metastatic disease and gave reference of Fig 1C. This is not correct it should be table 1. Even in the figure legend 1, line 91 states that the log-rank test p-value is shown and gives reference of Fig 1D, 1E. This is also incorrect as it relate to knockdown and overexpression of IF1.

 Following the reviewer’s request, in line 97 (now line 87) the alluded sentence has been amended to include also Table 1 as indicated. We apologize for the small size of the p value of the log-rank test shown in previous Fig. 1C, that seems to have confused the reviewer. In the revised version of Fig. 1C the p value (p<0.0004) is shown in bigger size to avoid confusion. In addition, in the revised version of the figure legend 1 we have incorporated the text of the statistical analysis of the overexpression and silencing of IF1 by western blotting and immunofluorescence, which were missing in the previous version, and could have contributed to confuse the reviewer in this point. We thank the reviewer for the comment.

The quality of Fig 2 is poor. The authors should work on it especially the heatmap where it is difficult to read. The authors could highlight only important genes.

 We apologize for the low quality of Figure 2. Following the reviewer’s request, we have improved the whole Figure, paying special attention to the heatmap, where we have increased the size of the lettering of the relevant genes identified in the IPA analysis.

Line 131 states the IPA analysis showed that the majority of activated pathways in shIF1 cells were related to cancer??

 Yes, the reason for our statement is, as we pointed out in the Discussion section (see lines 296-299), that the activation of p21-activated kinase, Ephrin receptor, p38MAPK, ERK/MAPK, HGF, PDGF and renal cell carcinoma signaling (Fig. 2F) are known to increase the aggressiveness of cancer. Following the reviewer’s comment, we have reformulated the sentence now reading “the majority of activated pathways in shIF1 cells are known to increase the aggressiveness of cancer”

Although the authors state the reason for comparison between shIF1 and IF1, the real comparison should be with their respective controls and not between knockdown and IF1. 

 This is an interesting comment, in fact, that was our original plan. However, when we noted that essentially the same genes were coming out in the two independent analysis (shIF1 vs CRL and IF1 vs CRL) when compared to the shIF1 vs IF1 comparison (albeit with less number of genes), we decided to continue the study as it is presented. In any case, the raw data of these comparisons are provided in the corresponding Supplemental Tables.

The cell growth assays in Figure 4A shows that shIF1 grew more than IF1 although significant; the quantification in 4B is marginal. Moreover if this is true, there is no difference between shIF1 and IF1 in vivo in xenograft models.

 As the reviewer pointed out, the growth rates of the cells in vitro were statistically different but the differences were low or of marginal relevance. This situation might explain per se why we observed no growth differences of the tumors in vivo. Apparently, as already discussed, the observation suggests that the metabolic differences noted between shIF1 and IF1 cells are irrelevant for the accretion of tumor mass. However, the differences are relevant in vivo when metastasis is taken into consideration, when the reliance of the cells on OXPHOS and the expression of cFLIP provides a clear advantage to shIF1 cells for colonizing the lung. As suggested by reviewer 1, a comment in this regard has been included in the revised version of the paper (See lines 306-308).

The authors found that c-FLIP mediated anoikis resistance in shIF1 cells. If this is the case then authors should test this in N and T samples of CRC patients and show that c-FLIP levels are reduced in tumors when they express IF-1 and show that knockout of c-FLIP in shIF1 cells lose their metastatic potential.

 We thank the reviewer for this interesting comment. In fact, following the reviewer’s suggestion we have studied the expression of cFLIP in the three patients of Fig. 1B and the results obtained (new Fig. 5G) show that the tissue content of cFLIP in normal colon is significantly higher than in the tumors, in partial agreement with reviewer’s suggestion. A sentence in this regard has been included in the revised version of the manuscript (see lines 229-231). Unfortunately, the N and T samples of the entire cohort of patients to analyze the expression of cFLIP in the carcinomas is not available any longer. However, we are already planning future studies to investigate the implication of cFLIP as a biomarker of cancer progression in colon and lung cancer patients.

The authors state that overexpression of IF1 in colon mitochondria promote the recruitment of anti-inflammatory immune cells in response to inflammation and refer to [29]. In that manuscript the authors describe about macrophages and not NK cells. So what was the rationale of looking at NK cells and not others like neutrophils or macrophages?  If NK cells play important role in tumor infiltration in spheroid models, the authors also show this relevance in CRC patients who have increased IF1 levels.

 The rationale at looking NK cells and not neutrophils or macrophages is based on the fact that NK cells are directly responsible of the cytolytic antitumor activity, which is the aim of this paper. Indeed, our future studies in cancer progression contemplate the study of tumor invasion by immune cells. In reference 29, we studied the role of mitochondrial IF1 in colon inflammation. Hence, in that study we analyzed the participation of macrophages and Tregs.

Minor points

Line 17 in abstract - less chances of disease recurrence, being IF1 (change to IF1 being) an independent predictor of survival

Has been modified as requested.

The reference 1 is old and should be updated with 2019 statistics

Reference 1 has been updated as requested, now reads Miller et al., (2019) CA Cancer J Clin.69, 363-385.

In panel 1C in the figure legend, the author states Kaplan-Meier curves for disease-free survival probability for the cohort of 38 colon cancer patients but in Fig they show only for 37 patients

 We apologize for the error; the text has been modified accordingly, now both read 37 patients.

Fig 1D the quantification is switched for knockdown and overexpression of IF1

We thank the reviewer for the observation and we apologize for the error. In the revised new Figure 1D, we have changed the position of the graphs.

The quantification for Fig 1E should include four groups of Ctrl, IF1, shC and shIF1

 We agree with the reviewer’s comment. In the revised new Figure 1E, we have incorporated the histogram to illustrate the comparison of shIF1 vs shC cells. The corresponding figure legend has been modified accordingly.

Round 2

Reviewer 3 Report

Major and Minor points

The author stated from previous comments that the activation of p21-activated kinase, Ephrin receptor, p38MAPK, ERK/MAPK, HGF signaling were activated in shIF1 cells than IF1 cells and showed that in Fig 2F. However, the authors didn’t find any changes in activation of these in shIF1 cells as shown in Fig S1 which is in contradiction to previous result section.

As stated previously, authors should test if chemokines responsible for recruitment of NK cells were decreased in shIF1 cells or other preliminary mechanism which could be follow-up study. For example CXCR3 or others in RNA-seq data which might support the NK cells recruitment and would strengthen the manuscript.

In line 82, the authors have missed the correction of number of CRC patients; it still says 38 CRC patients.

The authors should include the text of entire Fig 1 in one paragraph. For example, text in 2.1 and 2.2 could be combined into one paragraph with Figures 1A – 1C and Table 1 and title could be read as “Increased expression of IF1 is associated with increased survival of CRC patients”.

2.3 could be moved as 2.2 and Fig 1C and 1D could be included as Fig 2A and so on, just to get better flow for the reader or if space is an issue, Fig 1C and 1D could go as supplemental.

Similar could be done for Fig 5E - 5G